# COVID-19 Fear and Anxiety among Patients with Chronic Heart Failure: A Cross Sectional Study

**DOI:** 10.3390/jcm11216586

**Published:** 2022-11-07

**Authors:** Osama Alkouri, Yousef Khader, Issa M. Hweidi, Muntaha K. Gharaibeh, Mohamad Jarrah, Khaldoun M. Hamdan, Amina Al Marzouqi, Khaldoun Khamaiseh

**Affiliations:** 1Faculty of Nursing, Yarmouk University, Irbid 2116, Jordan; 2Department of Public Health, Faculty of Medicine, Jordan University of Science and Technology, Irbid 2116, Jordan; 3Faculty of Nursing, Jordan University of Science and Technology, Irbid 2116, Jordan; 4Faculty of Nursing, Al-Ahliyya Amman University, Amman 19328, Jordan; 5Department of Internal Medicine, Jordan University of Science and Technology, Irbid 2116, Jordan; 6College of Health Sciences, Health Services Administration University of Sharjah, Sharjah P.O. Box 27272, United Arab Emirates; 7Faculty of Medicine, Al-Balqa University, Al-Salt P.O. Box 19117, Jordan

**Keywords:** fear, anxiety, COVID-19, psychological distress, heart failure

## Abstract

Although the current management of COVID-19 is mainly focused on efficacious vaccine and infection control, the most common psychological reactions (such as fear and anxiety) associated with the COVID-19 pandemic have not been investigated and even neglected in patients with heart failure who are at greater risk for morbidity and mortality. We assessed COVID-19 related fear and anxiety among patients with heart failure and determined their associated factors. A cross sectional survey was conducted among 300 consecutive patients with heart failure during the period of March 2021–June 2021. Almost 50.7% of patients had fear of COVID-19 and 36.3% had coronavirus anxiety. Age > 55 was significantly associated with increased odds of fear (OR = 2.6) and anxiety (OR = 4.3). Patients with angina were more likely to have fear (OR = 3.0) and anxiety (OR = 2.2) and patients with chronic lung disease were more likely to have fear (OR = 3.0) and anxiety (OR = 3.3). Increased age, having angina, and having chronic lung disease were associated with increased odds of fear of COVID-19 and coronavirus anxiety. Psychological support needs to be integrated in patient care with special attention to physiological risk factors that are associated with COVID-19 comorbidities.

## 1. Introduction

The COVID-19 pandemic is a global health problem. As of 1 July 2022, 545,226,550 confirmed cases of COVID-19 have been reported including 633,4728 deaths [1]. COVID-19 is associated with increased morbidity and mortality [2]. Most countries have had escalating numbers of COVID-19 cases and deaths, placing a huge burden on their health care resources [3,4]. The devastating impact of COVID-19 has resulted in the financial and physical collapse of health care systems across the world, and reduced their ability to provide health care services [3]. In addition, COVID-19 can cause psychological reactions that can last longer than the pandemic itself [5,6,7,8]. The psychological impact of the virus, associated with increased morbidity, health care costs, and mortality, and its resonance in various contexts, is incalculable [8]. The most common psychological reactions associated with the COVID-19 pandemic include fear and anxiety [9,10,11]. Fear is defined as an unpleasant emotional state that is prompted when perceiving threatening stimuli [12,13], while anxiety is an emotion or future concern associated with intrusive thoughts and feelings of tension and worry [14,15]. 

Patients with chronic diseases, who often experience psychological reactions associated with COVID-19, are at greater risk of severe morbidity and mortality [16,17]. 

For example, heart failure can increase the risk of both catching the virus and experiencing its severe symptoms and complications [18]. Therefore, having chronic cardiovascular disease coupled with negative psychological reactions can cause the patient’s condition to deteriorate and exacerbate the symptoms. 

In Jordan, as of 24 June 2022, the number of confirmed cases of coronavirus has reached 1,696,937 [19]. The outbreak of COVID-19, coupled with the prevalence of cardiovascular diseases and its associated risk factors, has increased the severity of illness, and placed a further economic burden on Jordan’s health care system due to increasing hospitalization rates and health care expenditure [3,20]. 

Although the current management of COVID-19 is mainly focused on efficacious vaccine and infection control, the psychosocial domain has not been investigated and even neglected in patients with chronic conditions [21,22,23], even in Jordan. Therefore, this study aimed to assess fear and anxiety related to COVID-19 among patients with heart failure and determine their associated factors.

## 2. Materials and Methods

### 2.1. Study Design

A cross sectional survey study was conducted among 300 consecutive Jordanian patients with heart failure who attended outpatient clinics at a large university hospital in the north of Jordan during the period of March 2021–June 2021. The cross-sectional design can be utilized to assess behavior levels in a population with various socio-demographic characteristics [24,25]. The design has several advantages including cost-effectiveness, time saving and that can be tested in future studies, may assist with developing new theories [24,25]. Only Jordanian patients with native Arabic, older than 18 years, diagnosed with heart failure by a cardiologist as documented in their medical file were included in the study. Participants with pre-existing neurological or psychological disorders were excluded. 

### 2.2. Sample Size

The sample size was calculated using the Epicalc 2000. Assuming that 50% of patients have COVID-19 related fear or anxiety, the study requires a sample size of 196 to estimate the expected proportion with 7% absolute precision and 95% confidence. The sample size was increased to 300 to have sufficient power to test the association of COVID-19 related fear and anxiety with other independent factors [26]. 

### 2.3. Measurement Tools

The questionnaire package including a demographic sheet, the Fear of COVID-19 Scale, and the Coronavirus Anxiety Scale were used to collect the data. The demographic data collected were age, education, marital status, gender, and smoking. The clinical variables were hypertension, diabetes mellitus, angina, myocardial infarction, chronic lung disease, and high cholesterol. 

The forward and back translation method was used by an experienced bilingual professor to adapt the Fear of COVID-19 Scale and the Coronavirus Anxiety Scales in Jordan [27,28]. First, both scales were translated from English to Arabic, and second, they were independently translated back to English. Then, the study authors validated the coherence between the two versions. Additionally, both scales were piloted on twelve patients with heart failure (seven males and five females) who visited cardiac clinics and met the inclusion criteria for the study. The scales were found to be understandable, readable, clear, and culturally appropriate. 

The Fear of COVID-19 Scale is a five-point Likert scale used to assess fear associated with the COVID-19 pandemic. The scale comprises seven items relating to the emotional fear of COVID-19. A score of one indicates “strongly disagree”, and a score of five represents “strongly agree”. The range of total score was between 7 and 35. A higher score indicates a higher fear of COVID-19, while a lower score denotes a lower fear level. The scale was originally developed in the English language to assess fear associated with the COVID-19 pandemic [21]. The psychometric properties of the original scale including internal consistency (*α* = 0.82) and its test–retest reliability (ICC = 0.72) were acceptable [21,29]. Additionally, the original scale was translated to the Arabic language in a previous study and found to be reliable (Cronbach’s alpha coefficient of 0.88) [21,29]. In the present study, the piloted scale demonstrated high reliability, with a Cronbach’s alpha coefficient of 0.92. 

The Coronavirus Anxiety Scale is a 5-item mental health screener utilized to screen patients for COVID-19-related anxiety [30]. Patients are asked how often they have experienced anxiety symptoms over the last 2 weeks (0 ‘not at all’, 1 ‘rare, less than a day or two’, 2 ‘several days’, 3 ‘more than 7 days’, or 4 ‘nearly every day over the last 2 weeks’). The optimized cut-off score was ≥9 (90% sensitivity and 85% specificity) to distinguish between people affected with dysfunctional anxiety and those without. The scale has been found to be a highly valid and reliable [31,32]. The scale was previously translated to the Arabic language by Vally and Alowais [33]. In the present study, the translated scale also demonstrated high internal consistency, with a Cronbach’s alpha of 0.92. 

### 2.4. Data Collection Procedure

Ethical approval was obtained from the Institutional Review Board of the hospital where the study was performed, with an approval number of (103/136/2020). Two data collectors informed the administrators of the outpatients’ clinics about the study’s benefits and that participation would not harm/affect the patients’ treatment. Thereafter, the data collector met the general physician, who could access the patients’ medical records, to obtain an eligible patients list. On the cardiac clinic day, the data collectors debriefed the clinic nurse, regarding the objectives and ethics approval of the study, before the clinic started. Then, the clinic nurse notified eligible patients about the study to ensure voluntary participation. After obtaining the patients’ agreement, the cardiac nurse gathered the clinical information for each patient including hypertension, diabetes mellitus, angina, myocardial infarction, chronic lung disease, and cholesterol level, and provided it to the data collector. Then the participants were referred to a room within the cardiac clinic to meet the data collectors. Prior to commencing the data collection, the data collectors assured the participants that their responses would be confidential and anonymous, and that they were free to withdraw from the study at any time. Next, the participants signed a written informed consent form. Illiterate participants were allowed to participate if they had a family member or a friend to complete the questionnaires. 

### 2.5. Statistical Analysis

The data were analyzed using the Statistical Package of the Social Sciences Program (SPSS). Data were described using percentages, means, and standard deviation. The chi-square test was used to compare the prevalence rates of fear of COVID-19 and coronavirus anxiety between the categories of the studied factors. Binary logistic regression was used to determine the factors associated with the fear of COVID-19 and coronavirus anxiety. A *p*-value of 0.05 was considered statistically significant. 

## 3. Results

### Participants’ Characteristics

This study included 300 patients (172 males and 128 females) with chronic heart failure. Almost half of the patients were over 55 years of age (52.3%, *n* = 157). Of all patients, 48% had hypertension, 38.7% had diabetes, and 29.0% had myocardial infarction (MI). Table 1 shows their socio-demographic and clinical characteristics. 

## 4. Fear of COVID-19 and Coronavirus Anxiety 

Of all of the patients, 50.7% (*n* = 152) reported fear of COVID-19 and 36.3% (*n* = 109) had coronavirus anxiety. Fear of COVID-19 and coronavirus anxiety were significantly inter-correlated. The proportion of patients with coronavirus anxiety was significantly higher among patients with fear of COVID-19 compared to patients with no fear (68.4% vs. 3.4%, *p* < 0.001). 

The proportion of patients who experienced fear of COVID-19 and coronavirus anxiety varied significantly according to age, hypertension, angina, MI, and chronic lung disease (Table 2). The proportion of patients who experienced fear of COVID-19 was significantly higher among patients aged over 55 years compared to younger patients (66.9% vs. 32.9%), patients with hypertension (60.4% vs. 41.7%), angina (73.9% vs. 40.4%), myocardial infarction (65.5% vs. 44.6%), and chronic lung disease (73.1% vs. 46.0%).

The proportion of patients who had coronavirus anxiety was significantly higher among patients aged over 55 years compared to younger patients (53.5% vs. 17.5%), patients with hypertension (42.4% vs. 30.8%), angina (55.4% vs. 27.9%), myocardial infarction (47.1% vs. 31.9%), and chronic lung disease (61.5% vs. 31.0%). 

## 5. Multivariate Analysis of Factors Associated with Fear of COVID-19 and Coronavirus Anxiety

In the multivariate analysis, fear of COVID-19 and coronavirus anxiety were both associated with age, having angina, and having chronic lung disease (Table 3). Age > 55 was significantly associated with increased odds of fear (OR = 2.6) and anxiety (OR = 4.3). Patients with angina were more likely to have fear (OR = 3.0) and anxiety (OR = 2.2) and patients with chronic lung disease were more likely to experience fear (OR = 3.0) and anxiety (OR = 3.3).

## 6. Discussion

In the present study, about half (*n* = 152) of the sample had fear of COVID-19 (80 male; 72 female) and 36.3% (*n* = 109) had coronavirus anxiety (56 male; 53 female). This finding is in line with results from previous studies that reported a high prevalence of anxiety and fear related to COVID-19 [34,35]. The patients’ fear and anxiety can be attributed to the standard measures implemented by the Jordanian government for the control and prevention of health infection including social distancing, closure of airports and mosques, mask wearing, movement/travel restrictions, and curfew/lockdown [3,36]. Most people view quarantine as an unpleasant experience for several reasons including feelings of insecurity, social disconnection, and uncertainty regarding their health status as well as any potential complications in the future [34,35,37,38]. 

Furthermore, the psychological concerns can be attributed to the debilitating impact of COVID-19 on patients with heart failure [39], which has been identified as the most prevalent and serious complication related to the pandemic [40]. Heart failure can lead to the development of more severe COVID-19 symptoms and a higher death rate [39,41]. In a study of 113 Chinese patients who died due to COVID-19, 49% of them had heart failure and its complications [42]. Other studies, conducted in China, the USA, and Japan, have reported that heart failure represented 24% of the patients who had COVID-19 infection [41,42,43]. Importantly, heart failure was more common among patients who died with COVID-19 (52%) compared to survivors (12%) [44]. The deterioration of heart failure in COVID-19 infection can be attributed to direct invasion of the cardiac muscle by the virus, myocardial depression, and cardiomyopathy induced by an inflammatory response or stress [45,46]. 

Several physiological changes, associated with psychological distress during the pandemic, can predispose patients to or worsen cardiovascular disease including increased oxidative stress, catecholamine, cortisol levels, aggregation of platelet, inflammatory cells, endothelial dysfunction, and stimulation of the sympathetic nervous system [47,48]. Anxiety and fear are psychological disorders that can coexist with heart failure and other cardiovascular diseases, and are even considered major risk factors for cardiovascular disease such as coronary artery disease, stress-cardiomyopathy, arrhythmias, stress-cardiomyopathy, myocarditis, cardiogenic shock, and cardiac arrest [48,49]. 

This study showed that patients older than 55 years were more likely to be anxious and fearful of COVID-19 infection. Our results are in line with a study from Poland that found that the mean fear score among elderly patients aged over 60 years was 19.3 ± 5.6 (fear score ranged between 7 and 35) [50]. The concerns of older patients relating to COVID-19 might be explained by their knowledge that elderly people are vulnerable to COVID-19, with a high mortality rate. For example, in Italy, the death rate associated with COVID-19 infection in the elderly was 12.8% [50]. 

Older patients with heart failure are more likely to have other associated chronic comorbidities such as hypertension, diabetes, and chronic lung disease, which can increase the risk of severe morbidity and mortality related to COVID-19 [51]. Additionally, elderly patients with pre-existing health conditions are prescribed many types of medications, which can increase the anxiety levels associated with COVID-19. This was mainly found in patients taking cardiac medications [52]. In our study, many patients with cardiovascular diseases such as angina and/or myocardial infarction were prescribed anti coagulation therapy. A review study conducted in India indicates that patients receiving anti coagulation therapy experienced fears of catching the COVID-19 virus, which can cause abnormal alterations in blood clotting, and results in life threatening brain stroke as well as heart attacks [53]. Additionally, the deaths associated with COVID-19 are notably associated with pre-existing chronic diseases, particularly those common in the elderly such as diabetes and hypertension [54], and this fact can induce fear and anxiety [23]. 

Fear of COVID-19 in elderly patients can reduce the use of health care services, and eventually increase psychological distress. This was explained by a Polish study of Agrawal, Dróżdż, Makuch, Pietraszek, Sobieszczańska, and Mazur [51], which showed that fear of COVID-19 in elderly patients aged 60 and above (M = 67.9 ± 4.2) were associated with 10% cancelation of planned hospitalization. Another study from the United States showed that 33% of patients with HF, with the majority being elderly, indicated their hesitance to visit the hospital due to their fears of COVID-19 infection [52]. Thus, poor access to health care services can deteriorate the patient’s condition, increase the death risk, and ultimately trigger their fears and anxiety. The same study reported that mortality rate associated with COVID-19 is about five times higher in elderly patients aged 60 and over compared to young patients [51]. 

Information submitted about the COVID-19 outbreak and high number of daily confirmed cases can increase the levels of anxiety and fear, especially in elderly patients who spend many hours watching television and reading newspapers [55]. In Poland, a study of 500 elderly patients with chronic disease showed that 190 of them expressed that exposure to COVID-19 news was fearful and horrible [51]. Therefore, all news relating to the epidemiology of COVID-19, and its prevention and treatment should be unambiguous and not accompanied by disturbing images. Additionally, health care providers need to advise older patients to minimize their exposure to sensational news reporting of COVID-19, which may provoke the patients’ fear and stress and reduce their well-being through diverting their attention toward health promotion programs focused on health-related behaviors such as smoking cessation, healthy eating practices, and physical exercises [56,57,58].

The present study showed that the proportion of patients who had fear of COVID-19 and coronavirus anxiety were significantly higher in patients with hypertension, angina, myocardial infarction, and chronic lung disease. A study by Agrawal, Dróżdż, Makuch, Pietraszek, Sobieszczańska, and Mazur [51], showed that fear of COVID-19 was more prevalent in Polish patients with pre-existing chronic comorbidities such as heart failure, coronary heart disease, and chronic obstructive pulmonary disease COPD. This finding was also confirmed by a U.S. study by Kompaniyets et al. [59], performed on 540,667 patients with COVID-19, to identify frequent pre-existing comorbidities and their associated severe COVID-19 morbidity. Of those, 94.9% had at least one health condition such as hypertension (50.4%). Additionally, the greatest risk factors for mortality were diabetes plus its complication (aRR = 1.26; 95% CI, 1.24–1.28), anxiety and fear of health conditions (aRR = 1.28; 95% CI, 1.25–1.31) as well as the number of comorbidities. Interestingly, the National Center for Health Statistics of the U.S. also reported that 94% of mortalities related to COVID-19 were associated with other chronic comorbidities [60]. Results of various studies were also congruent with our finding [61,62]. 

One of the main concerns in patients living with a chronic disease is the delivery of medications [63,64]. A reduction in medical consultations, diagnostic tests, and treatments during the pandemic has been reported by patients and health care providers [65,66]. For example, a study from Spain showed that medication was cancelled in 90% of patients with COPD [67], which resulted in increased anxiety and fear and eventually worsened their health condition [67].

Hypertension and diabetes plus their complications (death, invasive mechanical ventilator, and ICU readmission) are associated with severe CIVID-19- related morbidity, due to their abnormal interference with inflammatory and hormonal pathways [59]. The high prevalence of these associated comorbidities in our study sample plus their considerable effect on COVID-19 can justify the patients’ anxiety and fear of COVID-19 [68]. 

For diabetes and hypertension, a recent systematic review of 1527 patients showed that the incidences of diabetes, cardio-cerebrovascular disease, and hypertension were 2- to 3-fold higher in severe cases admitted into ICU compared to no severe cases [69]. Specifically, diabetes and hypertension may increase susceptibility to sepsis, compromise immunity, and cause pancreatic infection [70], contributing to severe COVID-19 symptoms and increased mortality rates [71,72,73]. 

In patients with COPD, lung impairment, low-grade inflammation, and the chronic nature of this condition may weaken the immune system and ultimately increase the risk of lung infection [74]. Inadequate prevention and treatment may worsen COVID-19 symptoms such as tenacious dry cough, fatigue, intolerable difficulty breathing, and hypoxia. Development of pneumonia requires hospitalization for intensive care, and is associated with a significant chance of death [39]. In conclusion, the serious impact of the pandemic on patients living with COPD or any other pre-existing chronic conditions can rationalize their fear and anxiety. Regular assessment of the patients’ psychological status related to COVID-19, and careful management of pre-existing chronic diseases should be considered as a main component of heart failure management [47] to assist with the stratification of risks for severe illness [59]. Thus, frontline health care providers should use reliable media such as governmental television channels and COVID-19 emergency hotlines and valid screening tools to assess the effect of COVID-19 on psychological status such as the anxiety scale [74,75].

Implementation of informative campaigns can be beneficial for the prevention of COVID-19 spread and the reduction of psychological distress, particularly for elderly patients and those living with chronic conditions [51,59,76].

## 7. Limitations

This study had several limitations. First, the results yielded from the use of a self-reported questionnaire to assess the anxiety and fear of the participants may not provide a holistic image of the real situation. Second, the longitudinal impacts of COVID-19 on patients with heart failure could not be identified when using the cross-sectional research design. Finally, our study did not investigate the patients’ treatment regimens. 

## 8. Conclusions

Fear of COVID-19 and coronavirus anxiety are common among patients with heart failure. Increased age, having angina diabetes, and chronic lung disease were associated with increased odds of fear of COVID-19 and coronavirus anxiety. Early identification of people at high risk for or in the early stages of psychological distress, particularly the elderly and those living with chronic diseases, during this challenging time can enable health care providers to develop more effective health care interventions, services, and psycho education to alleviate psychological distress. The findings of the current study could also assist health care providers and health policy makers to develop interventions, especially for psychological support and self-management, which should be freely and easily accessible online for older patients with pre-existing comorbidities.

Elderly patients should use reliable sources such as government television channels and COVID-19 emergency hotlines to obtain information related to COVID-19 and reduce their exposure to media reports. The current study can be used as a foundation for longitudinal studies that investigate the long-lasting impacts of the COVID-19 pandemic on mental health. Furthermore, future studies using longitudinal designs could examine some of the mediating or moderating factors such as poorer health or greater media exposure, which may explain the association between age and greater fear/anxiety of COVID-19. 

## Figures and Tables

**Table 1 jcm-11-06586-t001:** The socio-demographic and clinical characteristics of patients with chronic heart failure (*n* = 300).

Variable	*n*	%
Gender		
Male	172	57.3
Female	128	42.7
Age		
≤55	143	47.7
>55	157	52.3
Educational level		
Illiterate	46	15.3
Low education	162	54.0
High education	92	30.7
Smoking	120	40.0
Marital status		
Married	226	75.3
Not married	74	24.7
Living alone	21	7.0
Hypertension	144	48.0
Diabetes mellitus	116	38.7
Angina	92	30.7
Myocardial infarction	87	29.0
Chronic lung disease	52	17.3
High cholesterol	108	36.0

**Table 2 jcm-11-06586-t002:** Fear of COVID-19 and coronavirus anxiety according to the socio-demographic and clinical characteristics.

	Fear of COVID-19					Coronavirus Anxiety				
	No		Yes			No		Yes		
	*n*	%	*n*	%	*p*-Value	*n*	%	*n*	%	*p*-Value
Gender					0.095					0.115
Male	92	53.5	80	46.5		116	67.4	56	32.6	
Female	56	43.8	72	56.3		75	58.6	53	41.4	
Age					0.000					0.000
≤55	96	67.1	47	32.9		118	82.5	25	17.5	
>55	52	33.1	105	66.9		73	46.5	84	53.5	
Educational level					0.344					0.051
Illiterate	19	41.3	27	58.7		22	47.8	24	52.2	
Low education	79	48.8	83	51.2		107	66.0	55	34.0	
High education	50	54.3	42	45.7		62	67.4	30	32.6	
Smoking					0.777					0.732
Yes	58	48.3	62	51.7		75	62.5	45	37.5	
No	90	50.0	90	50.0		116	64.4	64	35.6	
Marital status					0.348					0.154
Married	115	50.9	111	49.1		149	65.9	77	34.1	
Not married	33	44.6	41	55.4		42	56.8	32	43.2	
Living alone					0.871					0.767
Yes	10	47.6	11	52.4		14	66.7	7	33.3	
No	138	49.5	141	50.5		177	63.4	102	36.6	
Hypertension					0.001					0.037
Yes	57	39.6	87	60.4		83	57.6	61	42.4	
No	91	58.3	65	41.7		108	69.2	48	30.8	
Diabetes mellitus					0.051					0.342
Yes	49	42.2	67	57.8		70	60.3	46	39.7	
No	99	53.8	85	46.2		121	65.8	63	34.2	
Angina					0.000					0.000
Yes	24	26.1	68	73.9		41	44.6	51	55.4	
No	124	59.6	84	40.4		150	72.1	58	27.9	
Myocardial infarction					0.001					0.013
Yes	30	34.5	57	65.5		46	52.9	41	47.1	
No	118	55.4	95	44.6		145	68.1	68	31.9	
Chronic lung					0.000					0.000
Yes	14	26.9	38	73.1		20	38.5	32	61.5	
No	134	54.0	114	46.0		171	69.0	77	31.0	

**Table 3 jcm-11-06586-t003:** Multivariate analysis of factors associated with fear of COVID-19 and coronavirus anxiety.

Variable	Fear of COVID-19				Coronavirus Anxiety			
	OR	95% Confidence Interval		*p*-Value	OR	95% Confidence Interval		*p*-Value
Age (>55 vs. ≤55)	2.6	1.5	4.5	0.001	4.3	2.4	7.9	<0.001
Hypertension	1.6	0.9	2.7	0.113	1.0	0.6	1.8	0.989
Angina	3.0	1.7	5.4	<0.001	2.2	1.2	3.9	0.008
Myocardial infarction	1.5	0.8	2.6	0.199	1.2	0.7	2.2	0.495
Chronic lung	3.0	1.5	6.1	0.003	3.3	1.7	6.5	0.001

## Data Availability

Data are available upon reasonable request from the corresponding author.

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
