# Peer review of "COVID-19 Fear and Anxiety among Patients with Chronic Heart Failure: A Cross Sectional Study"

_jcm, 2022, doi:10.3390/jcm11216586_

Round 1

Reviewer 1 Report (Previous Reviewer 3)

Better written than previous version, although more effort needs to be put into editing the manuscript. 

Reviewer 2 Report (Previous Reviewer 1)

The authors have responded adequately to the issues raised for review.
Therefore, this article has been substantially improved and can be assessed by the relevant Editor,
Best regards and thank you for your confidence.

This manuscript is a resubmission of an earlier submission. The following is a list of the peer review reports and author responses from that submission.

Round 1

Reviewer 1 Report

I am grateful to be able to review this research, which I believe to be interesting and of high quality. In addition, it has been carried out by a multidisciplinary team, and I suggest that in order to improve it, a series of observations should be taken into account, which I indicate below:

- It would be advisable in this case and at the end of the document to indicate the number/reference of the Ethics Committee. Also, as the research was carried out in a single hospital, equal opportunities should be taken into account, as it is indicated how the information on illiterate patients is collected, but it does not include patients who do not understand the language (what was included, was an interpreter needed, etc.)?

- line 186: the study would be more interesting if it had indicated the number of women and men who suffer from a pathology with respect to the variables explored, above all to present in tables 2 and 3. Also to indicate where the studies are carried out is indicated throughout the discussion, as in the first part, sometimes these do not appear.

- line 192-194: this paragraph would benefit from more bibliography.

- line 199: indicate the place where the study was carried out.

- line 225-229: also indicate the location of the study.

- line 233-234: studies with older patients are presented, but it would be convenient to present the advanced age of the sample.

- line 246-248: indicate how the health professionals "advise"... it is not very specific.

- line 303: what are the reliable sources proposed?

???

- line 307: it is advised, be another point, and not included in the discussion.

The end of the discussion seems more appropriate to include it in the conclusion.

In the Conclusion, we find the diseases included as variables, but diabetes does not appear, is this correct? 

Reviewer 2 Report

I really appreciate the opportunity to review this manuscript entitled “COVID-19 fear and anxiety among patients with chronic heart failure: A cross sectional 2 study”. This is important to assess fear and anxiety related to COVID-19.  I only remark some issues (most of them in methods) in order to improve the quality of this manuscript.

The abstract is clear but it is important to explain the design, why authors chose this kind of patients and not general population?

Introduction was well structure and shows the necessity for this research. At the methods section, there are some questions that should be review. If this study, how authors assure that patients do not have previous psychological disorders? I also do not understand why parents were divided according to the age 55. Why variable age was categorized dichotomously?

Results were clear. Discussion summarize and explain in a good way the finding but, from my point of view it would be interesting to discuss about age differences, what is consider elderly people? And what about differences between educational status? From my point of view discussion about diabetes and hypertension in too extend. Which are the future lines of research?

Conclusions were correct.

Be careful with the change of letter size in the citations.

Reviewer 3 Report

I think the authors should explain how fear and anxiety are different constructs. Fear is the response to a well-know threat and anxiety is more diffuse and a response to a less well-defined threat. Although fear and anxiety are related, they are not the same. In the introduction, the authors should present this and let the readers know they are different but related constructs. 

I think that the manuscript should be extensively edited to make it readable. For example, at the bottom of page 6, the sentence should read "the data collectors assured the participants that their responses would be confidential and anonymous, and that they were free to withdraw from the study at any time".  Many times throughout the manuscript, words were omitted, tenses changed unpredictably, and sentences used odd phrasing. For example, on page 7, the authors should say' Almost half of the patients were over 55 years of age". On this same page, the authors don't present the p value (I assume its 0.05) and simply say 'A p-value was considered statistically significant". 

I think the discussion should be rewritten to make it more readable. It doesn't seem very well organised and appears to simply list a series of disjointed facts. Although these points should be discussed, they need to be connected in a more convincing narrative.

Although this is a cross-sectional study, the authors should state that future studies using longitudinal designs could examine some of the mediating or moderating factors, such as poorer health or greater media exposure, that may explain the association between age and greater fear/anxiety of COVID. 

Round 2

Reviewer 3 Report

The authors have addressed my concerns and improved the manuscript. However, additional proofreading is needed. For example, on page 13 the authors use the phrase 'self-preferable programs'. This doesn't make sense to me. In the abstract, the authors state 'Psychological support is needed to be integrated in patient care with special attention to physiological risk factors which is associated with COVID-19 comorbidities.' It should read 'needs' rather than 'is needed' and since factors are plural, it should read 'which are associated'. There are many other places where the standard of English needs to be improved, but I cannot list every instance for the entire manuscript.